# Exploring the role of temperature and other environmental factors in West Nile virus incidence and prediction in California counties from 2017–2022 using a zero-inflated model

**Noah Parker**⊙*

Department of Epidemiology, Berkeley School of Public Health, University of California, Berkeley, Berkeley, California, United States of America

* noahtparker3@berkeley.edu

## Abstract

West Nile virus (WNV) is the most common mosquito-borne disease in the United States, resulting in hundreds of reported cases yearly in California alone. The transmission cycle occurs mostly in birds and mosquitoes, making meteorological conditions, such as temperature, especially important to transmission characteristics. Given that future increases in temperature are all but inevitable due to worldwide climate change, determining associations between temperature and WNV incidence in humans, as well as making predictions on future cases, are important to public health agencies in California. Using surveillance data from the California Department of Public Health (CDPH), meteorological data from the National Oceanic and Atmospheric Administration (NOAA), and vector and host data from VectorSurv, we created GEE autoregressive and zero-inflated regression models to determine the role of temperature and other environmental factors in WNV incidence and predictions. An increase in temperature was found to be associated with an increase in incidence in 11 high-burden Californian counties between 2017–2022 (IRR = 1.06), holding location, time of year, and rainfall constant. A hypothetical increase of two degrees Fahrenheit—predicted for California by 2040—would have resulted in upwards of 20 excess cases per year during our study period. Using 2017–2021 as a training set, meteorological and host/vector data were able to closely predict 2022 incidence, though the models did overestimate the peak number of cases. The zero-inflated model closely predicted the low number of cases in winter months but performed worse than the GEE model during high-transmission periods. These findings suggests that climate change will, and may be already, altering transmission dynamics and incidence of WNV in California, and provides tools to help predict incidence into the future.

## Author summary

West Nile Virus is a disease that is spread by mosquitoes. Though it commonly infects birds, transmission to humans is possible and can lead to severe health effects.

**Data Availability Statement:** Limited, de-identified human West Nile virus surveillance data are publicly available via the California Health and

Human Services Open Data Portal (https://data.chhs.ca.gov/dataset/west-nile-virus-cases-2006-present). Readers can request access from the Infectious Diseases Branch (https://data.chhs.ca.gov/dataset/west-nile-virus-cases-2006-present) or by request from the Infectious Diseases Branch (https://www.cdph.ca.gov/CDPH%20Document%20Library/ControlledForms/cdph9078.pdf). Mosquito and bird infection data is available through request from VectorSurv/CalSurv at https://vectorsurv.org (https://vectorsurv.org/assets/files/calsurv_data_policy.pdf). Temperature and precipitation data is publicly available via the National Oceanic and Atmospheric Administration website (https://www.weather.gov/wrh/Climate?wfo=hnx).

**Funding:** The author(s) received no specific funding for this work.

**Competing interests:** The authors have declared that no competing interests exist.

Temperature is known to affect the transmission cycle of West Nile virus, but it is unclear how global warming might change who, or how many people, may get infected with the virus. In this study, the researchers looked at how climate change may affect West Nile virus in California, and how health officials may better be able to predict future cases. The study found that there could be an increase in West Nile virus cases in humans due to increases in temperature in the next 20 years, but that we already have many tools and sources of data to predict cases. These findings reinforce the possible consequences of climate change on human health, and aid in the understanding in the complex relationship between climate and infectious diseases.

## Introduction

West Nile virus (WNV) is a vector-borne flavivirus that is responsible for the most infections of any mosquito-transmitted disease in the United States [1]. West Nile virus was first detected in Southern California in 2003, before quickly spreading to all 58 Californian counties within a year [2]. California bears the largest burden of WNV infections, with 4,035 of the nation's 21,869 infections (18%) between 2009 and 2018 being reported in the state [3]. Only 20% of infected people will show any symptoms, while less than 1% will show neurological symptoms, manifesting as West Nile neuroinvasive disease (WNND) [3]. Despite the low symptomatic and severity rate, California reported 326 fatalities due to WNND between 2003 and 2018, as the fatality rate of WNND is about 10% [4].

Many environmental factors are known to impact WNV transmission, with most transmission occurring in non-human hosts. Humans become infected through the bite of a female mosquito from the *Culex* genus, though WNV cannot be spread between or from humans—they are a dead-end host [3]. The enzootic cycle mainly flows through mosquitoes and certain bird species, such as corvids, finches, and sparrows, though infections of horses and chickens are also common [3]. Given that WNV incidence in mosquitoes, bird, horses, and other species are an indicator of increased transmission in the enzootic cycle, high wildlife prevalence can serve as a good predictor of the transmission risk to humans [4]. Additionally, certain meteorological conditions, such as temperature and rainfall, play an important role in mosquito spawning and behavior and have been shown to be positively associated with human WNV infections [5]. In particular, temperature has been shown to affect mosquito life cycle traits and interactions with pathogens, in turn affecting WNV transmission [2,6,7]. Prior research has highlighted the predictive power of these environmental factors on West Nile cases in the US and Canada, with the possibility that prediction models might be applied in other higher-burdened areas [1,8].

While other studies have modeled the importance of temperature and certain other environmental factors in WNV transmission in California [9], as well as the ability of the state's warning system to determine outbreaks [4], there is sparse research that has attempted to quantify excess cases that may be associated with changing meteorological conditions. Few studies also expand further to create prediction models using currently collected data sources. This analysis is especially important given the projected rise in temperature in California due to worldwide climate change and can help state officials plan for possible increases in future WNV transmission. To our knowledge, no other papers have used a zero-inflated regression model to examine WNV in California. A zero-inflated model can be applied in cases where there are many expected zeroes in a dataset. In the context of West Nile virus, a zero-inflated regression model may have advantages over traditional models given the clear seasonality of

human infections and the near certainty of zero cases for much of the year. The effectiveness of such a model over a negative binomial regression model that is more traditionally used in longitudinal infectious disease analysis is explored in this paper.

In our paper, we use a zero-inflated regression model and a generalized estimating equation (GEE) that incorporates autoregressive structure to measure the association between temperature and WNV incidence, determine the number of excess cases that would result from an increase in temperature, as well as compare the predictions each make for future WNV transmission in California. We hypothesize that temperature will be positively associated with WNV in California after adjusting for rainfall, location, and month of year; meanwhile, meteorological conditions and environmental factors, such as infected dead birds and mosquito pool positivity rate, will be able to predict future WNV cases. We also hypothesize that the zero-inflated model will provide more accurate and precise predictions than the GEE model.

## Methods

### Data

West Nile virus is a national and state notifiable condition [10]. As such, case data was collected by the California Department of Public Health as part of their surveillance program. While weekly case incidence is reported, monthly incidence was used in the study to remove inconsistent lags in reporting. Only counties with a population over 250,000 residents and a total of 40 or more cases over the six-year period (2017–2022) were considered high-burden counties and included in the analysis. This totaled 11 counties: Fresno, Kern, Los Angeles, Merced, Orange, Riverside, Sacramento, San Bernadino, San Joaquin, Stanislaus, and Tulare. A restriction was placed on cases due to most counties having a small or zero case count of WNV over the study period, while a minimum county size was enacted to protect cases' identities in small counties. We decided to only use the most recent six-year period of case data, as this reflected a tradeoff between having a large number of observations while maintaining a sufficiently contained time range where transmission and meteorological conditions expressed similar characteristics. Average monthly temperature across California rose sharply in the early 2010's, before peaking in 2015 [11]. Temperatures have since remained higher than 20th century averages, but have stayed relatively stable below the peak, fulfilling the assumption that meteorological conditions are expressing stable characteristics during our study period [11]. We also ran a sensitivity analysis using different sets of years in our data set—three sets of data using five years and one set of data using four years—to determine if the specific years used in the analysis influenced the strength and direction of the association of interest. All the sets produced very similar estimates of the incidence rate ratio between prior month temperature and incidence, leading us to have confidence in the chosen time period.

Average daily temperature (aggregated across the month in Fahrenheit) and rainfall (summed for the month in inches) were collected from the National Oceanic and Atmospheric Administration. Mosquito pool positivity rates (percentage of mosquito pools within a county that tested positive for WNV) and bird infection counts (count of dead birds that tested positive for WNV) were obtained through California's VectorSurv program under request number 000067.

### Zero-Inflated and GEE models for measure of association

**The models and variables.** Negative binomial regression was conducted in both models instead of Poisson because the case data had an overdispersed distribution. Due to the time it takes for a person to become sick enough to seek treatment, as well as the reporting delays of cases to the state health department, we assumed that meteorological conditions and reported

case counts are not directly related in time. Given that substantial lags between case onset and case reporting—an average of 5 weeks—have been found in other WNV outbreaks, temperature and rainfall were both lagged one month, so that the variables *prior month temperature* and *prior month rainfall* would better reflect the current month's cases [12]. As the reporting of vector data are also subject to delays in reporting and biological processes, it was assumed that the mosquito pool positivity rate and dead bird count follow similar delays as cases and therefore were not lagged [12]. This selective lagging strategy has been used in other work that models WNV [1]. Rainfall, month of year, and county were adjusted for as confounders in the relationship between temperature and WNV cases. Controlling for these variables also helps control for the seasonality and location dependence of West Nile infections. The percentage of positive mosquito pools and positive dead birds were determined to be on the causal pathway and were not analyzed in the models testing for a measure of association, the incidence rate ratio.

**Zero-inflated.**   We used two different methods to estimate an association between temperature and West Nile virus incidence. First, we utilized a zero-inflated negative binomial regression model to model the count of cases in the 11 high-burden counties. This type of model is useful in situations where there are many excess zeroes in a distribution of count data. The zero-inflated model has two distinct processes—one to model the excess zeroes and the other to model the count [13,14]. This removes some of the noise in the association created by the zero counts, and might provide a more accurate association in the observations that have non-zero counts. This method was used to account for the large number of months with zero cases in our data, as there is little WNV transmission outside of June through November. 525 of the monthly case counts were zero, representing 66.3% of the observations.

We represent the number of cases as $y_{it}$, where *i* represents the county (i = 1. . ..11) and *t* represents the month of observation over the six-year period (t = 1. . .72). We assume that structural zeroes appear in the data at time *t* and county *i* at a probability of $\pi$. Therefore, there are two processes in which data can be generated for each observation: the first, which generates a case count of 0 with probability of $\pi$; the second, which generates a case count corresponding to a negative binomial model, represented by $g(y_{it}|X_{it})$ with a probability of $1-\pi$.

$$y_{it} \sim \begin{cases} 0 & \text{with probability} & \pi_i \\ g(y_{it}|X_{it}) & \text{with probability} & 1 - \pi_i \end{cases}$$

Therefore, we assume the probability that the expected number of cases, $Y_{it}$, is equal to $y_{it}$ is given by the zero-inflated negative binomial model below. $X_{it}$ represents the vector of covariates in the model, while $Month_{it}$ expresses the calendar month at county *i* and time *t*, which is believed to well predict whether there will be cases in a given observation or not.

$$\Pr(Y_{it} = y_{it}|X_{it}, Month_{it}) = \begin{cases} \pi(Month_{it}) + \{1 - \pi(Month_{it})\}(g(0|X_{it})) & \text{if } y_{it} = 0 \\ \{1 - \pi(Month_{it})\}(g(y_{it}|X_{it})) & \text{if } y_{it} > 0 \end{cases}$$

The function representing the negative binomial model, $g(y_{it}|X_{it})$, is defined below, where the vector of covariates in the model, $X_{it}$, includes the prior month temperature, prior month rainfall, the county, and the month. Prior month temperature is the exposure of interest on the outcome, while prior month rainfall, county, and month are all confounders of this relationship. While county and month are shown here as being expressed by one $\beta eta$ term, they are analyzed as factor variables in the model, meaning that a dummy variable was created for each category level (10 for county and 11 for month), and a $\beta eta$ term was calculated for each.

$$g(y_{it}X_{it}) = E(y_{it}X_{it}) = (exp^{X_{it}\beta})$$

$$g(y_{it}|X_{it}) = E(y_{it}|X_{it}) = \exp[\beta_0 + \beta_1(PriorMonthTemp) + \beta_2(PriorMonthRainfall) + \beta_3(County) + \beta_4(Month)]$$

**Negative binomial GEE.** Second, we used a negative binomial generalized estimating equation model with autocorrelation features as a comparison to the zero-inflated model. This type of model is more commonly used in longitudinal infectious disease analysis, as the ability to imply a correlation between observations more accurately reflects real-world dynamic of disease transmission. However, unlike the zero-inflated model, each observation contributes to the association—there is no consideration for zero versus non-zero case count [15].

We utilized an "ar1" auto-correlation structure, meaning cases in one month were strongly correlated with cases in the month before and the month after, but not other months. This structure is thought to accurately describe infection dynamics in our system, as the cases in county $i$ at time $t$ are dictated by how many mosquito infections, and subsequent human infections, were present at time $t-1$, as well as influences how many cases there will be at time $t+1$ given the same dynamics. The model follows the same form as the negative binomial part of the zero-inflated model, but has an added autoregressive term, $\sigma$, that follows a distribution of "ar1", as shown in the model below.

$$\log[E(Y_{it}|X_{it})] = X_{it}\beta$$

$$\log[E(Y_{it}|X_{it})] = \beta_0 + \beta_1(PriorMonthTemp) + \beta_2(PriorMonthRainfall) + \beta_3(County) + \beta_4(Month) + \sigma$$

$$\sigma \sim AR(1)$$

**Standardization.** Model standardization was conducted to retrieve a risk difference (in the number of excess cases) between a hypothetical dataset, where temperature was increased by 1, 1.5, 2, 2.5, or 3 degrees Fahrenheit (with all other variables kept constant), and the observed dataset. We created a population where *prior month temperature* was increased by these set amounts across all observations, later using it to fit a zero-inflated model and find the total number of cases expected in the six-year study. The risk difference is the total number of cases expected in the hypothetical population minus the number of cases expected given the observed data. While this risk difference cannot be interpreted as an association between temperature and West Nile incidence—due to it being a hypothetical population—it can provide insight into possible repercussions of a changing climate in the future. Bootstrapped sampling, using the bias-corrected and accelerated technique (BCA), was subsequently applied to measure the uncertainty in our risk difference calculations.

## Prediction models

Both the zero-inflated model and autoregressive negative binomial model were used as prediction models for WNV cases in 2022. Both models were trained on the data from 2017–2021 and then tested on the data from 2022. The models followed the same assumptions and equations as the models specified above, respectively, but were made to include different sets of the covariates in the data. Since they are prediction models, all variables, including percentage of positive mosquito pools and positive dead birds—which were left out of the measure of association models due to existing on the causal pathway between temperature and WNV incidence—were included in the analysis. The four different prediction sets tested were: "All," comprising all the covariables listed; "Host/Vector," comprising dead infected birds and mosquito

positivity rate; "Just Temperature," comprising just prior month temperature; and "Weather," comprising prior month temperature and prior month rainfall. We opted to use mean squared error (MSE) to compare the models over information criteria tests, such as the AIC or BIC, as we didn't have concrete likelihood estimates from the GEE model [16]. Though mean squared error doesn't account for and penalize the number of predictor variables in each model, the lack of a large difference in variables used (4 vs 2 vs 2 vs 1 in "All," "Host/Vector," "Weather," and "Just Temperature," respectively) and a clear difference in each model's mean squared error highlighted a clear best fit for the data.

## Ethical dimensions

Given the ecological design of the study and the passively collected surveillance case data, no data was actively collected on human subjects. No identifiers were included in the data, and only aggregated data from high-burden, highly populated counties were used in the analysis to protect individual's identities. Therefore, there was no risk to any individuals who may have been included in the data. While this analysis was conducted after all of the infections and any findings will not be of benefit to the individual, the results may still benefit these communities in the future.

## Results

### Descriptive analysis

Fig 1 plots the mean temperature and West Nile virus cases across all 11 high-burden counties included in the study. This figure shows the clear seasonality of WNV activity, with the peak of each year occurring in the summer, when the temperature is highest. This also highlights the almost complete lack of cases in the colder winter months. There is high variability in the number of cases in these counties in a single year: there were 515 cases reported in 2017, while just 77 in 2021. The analysis of detangling seasonality and temperature to examine the association between temperature and WNV cases is presented below.

### Statistical models and standardization

Both the zero-inflated and GEE with autoregressive correlation models show a positive correlation between prior month temperature and WNV cases in a month (Fig 2). Each model was adjusted for the variables prior month rainfall, county, and month, which were determined to be confounders in the relationship of interest.

Both models had nearly the same estimation of 1.05 for the incidence rate ratio—this represents that a one degree increase in prior month temperature is associated with a 1.05 times greater incidence of West Nile virus in a month, controlling for prior month rainfall, location, and the time of year. The IRR estimates from both models were statistically significant, though the autoregressive GEE model had a narrower 95% confidence interval (Fig 2).

The models can also be extrapolated to larger changes in temperatures, as is predicted by some climatologists [17–19]. Some estimates suggest that California's average temperatures may rise by two degrees Fahrenheit by 2040 [17,18]. Given a two degree increase in temperature, and holding all other variables constant, the zero-inflated models suggest that this would be associated with a 1.12 times greater incidence rate, while the GEE model reports this number to be 1.10.

Fig 3 shows the risk difference of WNV cases conducted through model standardization, comparing the predicted cases under the observed data versus a hypothetical dataset where all temperature observations were increased by two degrees Fahrenheit. As shown in the figure, a two degree increase in temperature, holding all other variables constant, results in more cases (a positive risk difference) in the zero-inflated model. This increase is heavily concentrated in summer

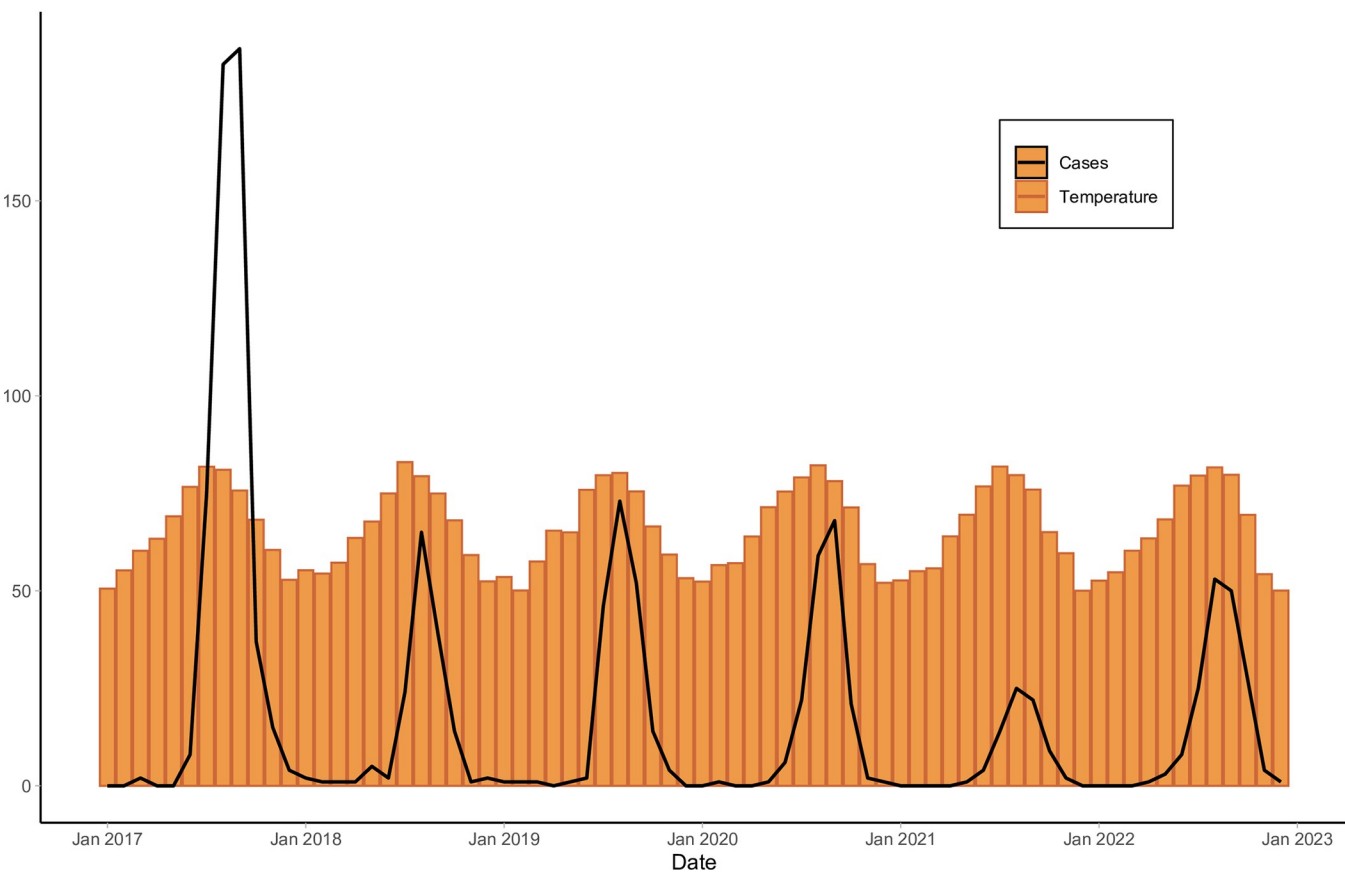

**Fig 1. Average monthly temperature and total monthly West Nile virus cases in 11 high-burden California counties.**

months, with at least eight excess cases predicted every year during peak months. An increase in temperature barely affects winter months where there is little to no transmission. Across the six-year study period, the risk difference between the hypothetical data, with a two degree temperature increase, and the observed data is 148.7 (95% confidence interval of 28.8–292.8) cases for the zero-inflated model. This averages out to 24.8 (4.8–48.8) excess cases per year.

The number of excess cases has a positive linear relationship with the level of temperature increase, as shown in Fig 4. Increases of all amounts, even at one degree higher temperature, showed a significant positive risk difference. The amount of uncertainty also increases with

**Assocation Between Incidence of WNV and Prior Month Temperature in High-Burden California Counties**

| Characteristic | Zero-Inflated Regression | | GEE with Autoregressive Correlation | |
|---|---|---|---|---|
| | IRR[1,2] | 95% CI[2] | IRR[1,2] | 95% CI[2] |
| Prior Month Temperature | 1.057 | 1.013, 1.103 | 1.048 | 1.012, 1.086 |

[1] Adjusted for Prior Month Rainfall, County, and Month

[2] IRR = Incidence Rate Ratio, CI = Confidence Interval

**Fig 2. Association between incidence of WNV and prior month temperature in our study population.**

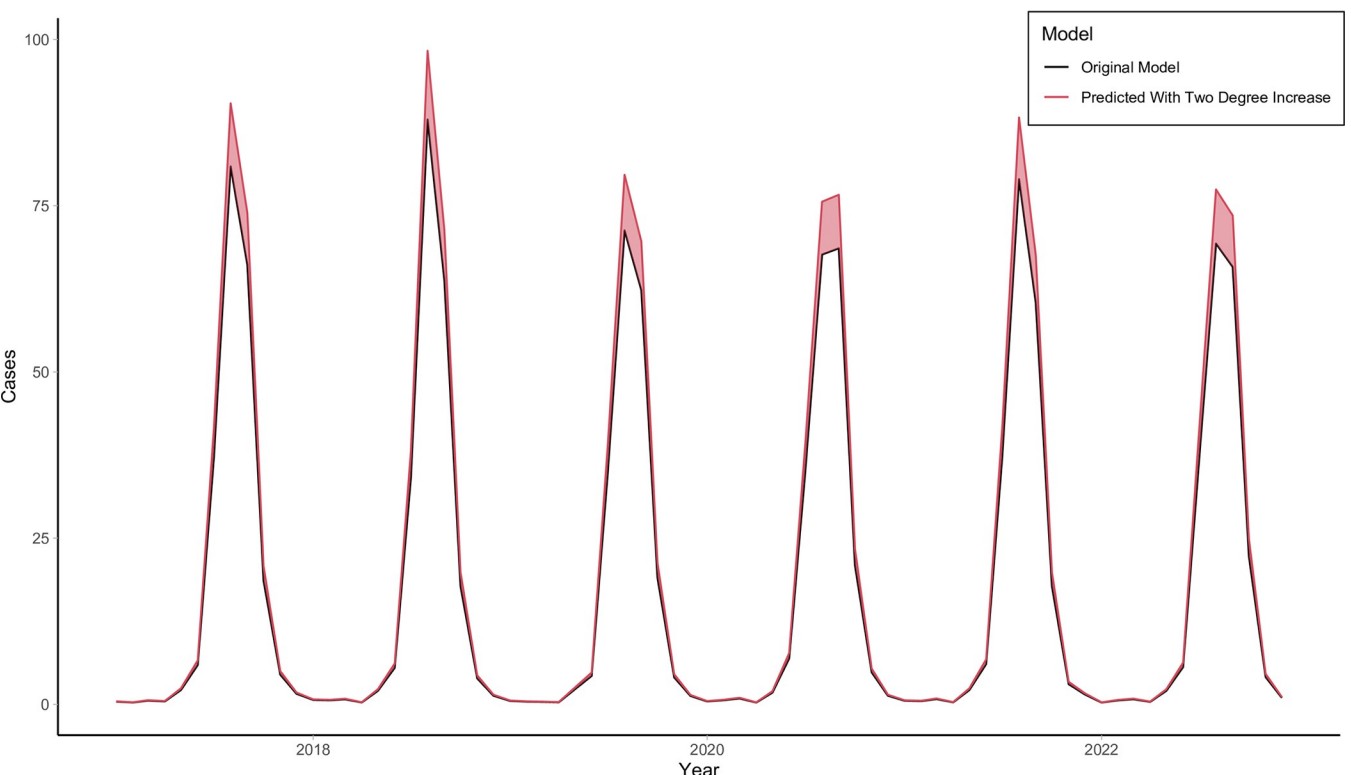

**Fig 3. Risk difference (shaded red) between original zero-inflated model and two degree increased zero-inflated model.**

temperature; there are no significant differences in the number of excess cases across any of the temperature increases.

## Prediction models

Fig 5 shows the output of the prediction model with all variables for the zero-inflated model and GEE model. As in the measure of association model, both types of models showed fairly similar results, as prior month temperature, mosquito pool positivity rate, and dead birds positively predicted WNV cases in each model, while prior month rainfall negatively predicted WNV cases. The models' predictions slightly differed in their estimates of the beta for each variable, and while small in absolute size, the differences were significant, as seen in later analysis.

Fig 6 plots the predicted cases in the 11-high burden counties in 2022 for each variable set —across both model types—against the actual number of reported cases, as well as each model's MSE. All variable sets of the zero-inflated model fit the actual data well during zero case count months (November-June) but over-estimated the number of cases in the peak summer months. "Just Temperature" and "Weather" both over-estimated the peak summer months and incorrectly predicted the shape of the incidence curve. It's possible that since the zero-inflated model already uses the month of year, which is very strongly tied to weather patterns, to determine non-zero case counts, adding weather data to the prediction model did not increase the strength of the fit. "All" variables provided the best fit—the lowest MSE—to the data out of the zero-inflated subset.

The GEE subset of models showed more similar fits, again most likely due to the model not having the month predictor for zero case counts. In contrast to the zero-inflated models, the

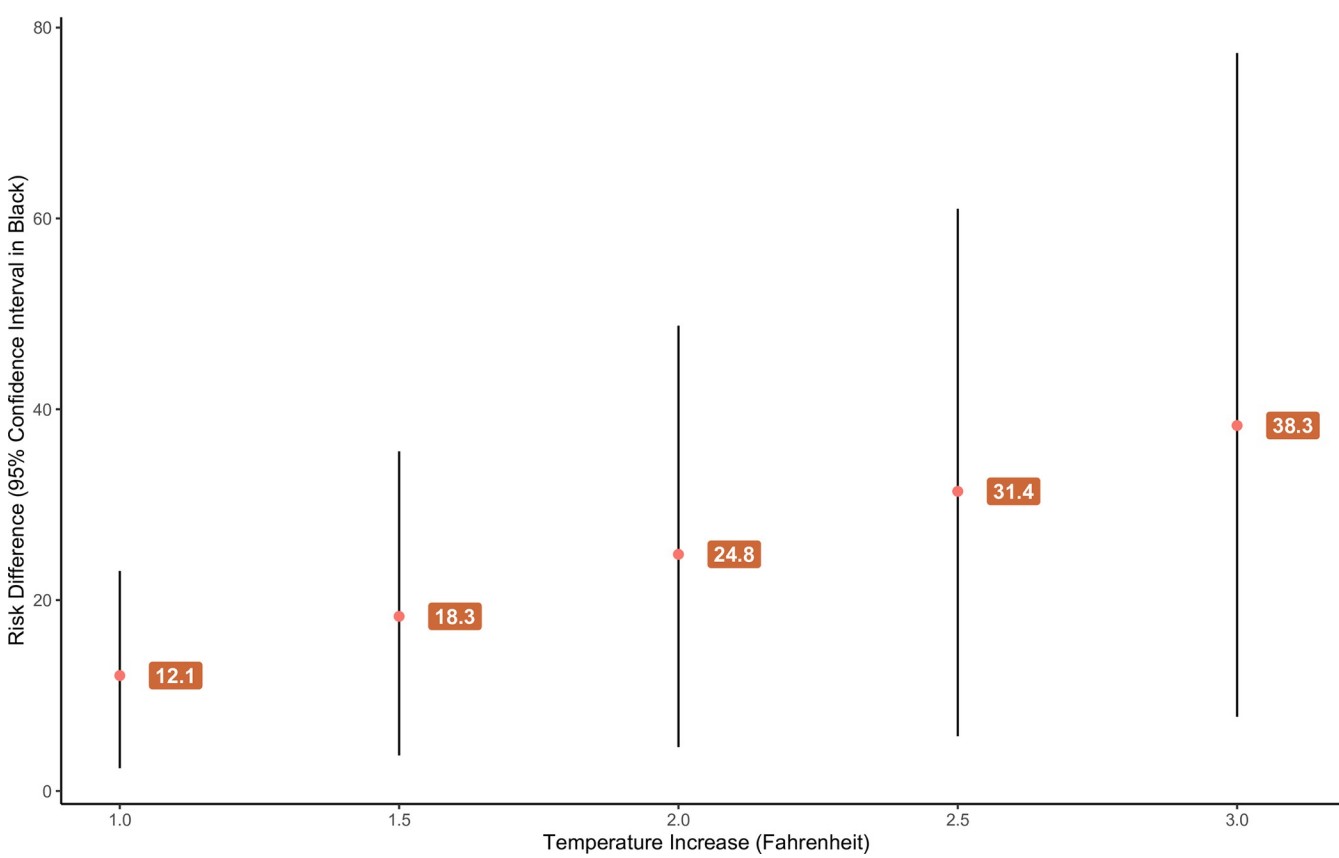

**Fig 4. Risk difference between original zero-inflated model and zero-inflated model with various temperature increases.** Bootstrapped 95% confidence interval in black.

GEE prediction models followed the peak of cases in August much more closely, though it also overestimated the actual reported number of cases slightly. The GEE model overestimated the number of cases in months with low case counts, however, often at scales of three to four times larger. The "Host/Vector" model especially produced a skewed fit, as even with a mean squared error of 78.2, it predicted 10 cases a month in the winter months and overall had a non-normal shape. The "All" GEE model resulted in the best fit across all of the prediction models with a mean squared error of 17 and a closely following incidence curve.

Prediction Model Output Between Environmental Factors and WNV Incidence

| Characteristic | Zero-Inflated Regression | | GEE with Autoregressive Correlation | |
| --- | --- | --- | --- | --- |
| | Beta | 95% CI[1] | Beta | 95% CI[1] |
| Prior Month Temperature | 0.08 | 0.04, 0.11 | 0.07 | 0.02, 0.12 |
| Prior Month Rainfall | -0.21 | -0.48, 0.07 | -0.24 | -0.38, -0.11 |
| Mosquito Positivity Rate | 0.05 | 0.03, 0.06 | 0.04 | 0.02, 0.05 |
| Dead Birds | 0.06 | 0.04, 0.08 | 0.02 | 0.01, 0.04 |

[1] CI = Confidence Interval

**Fig 5. Prediction model outputs.**

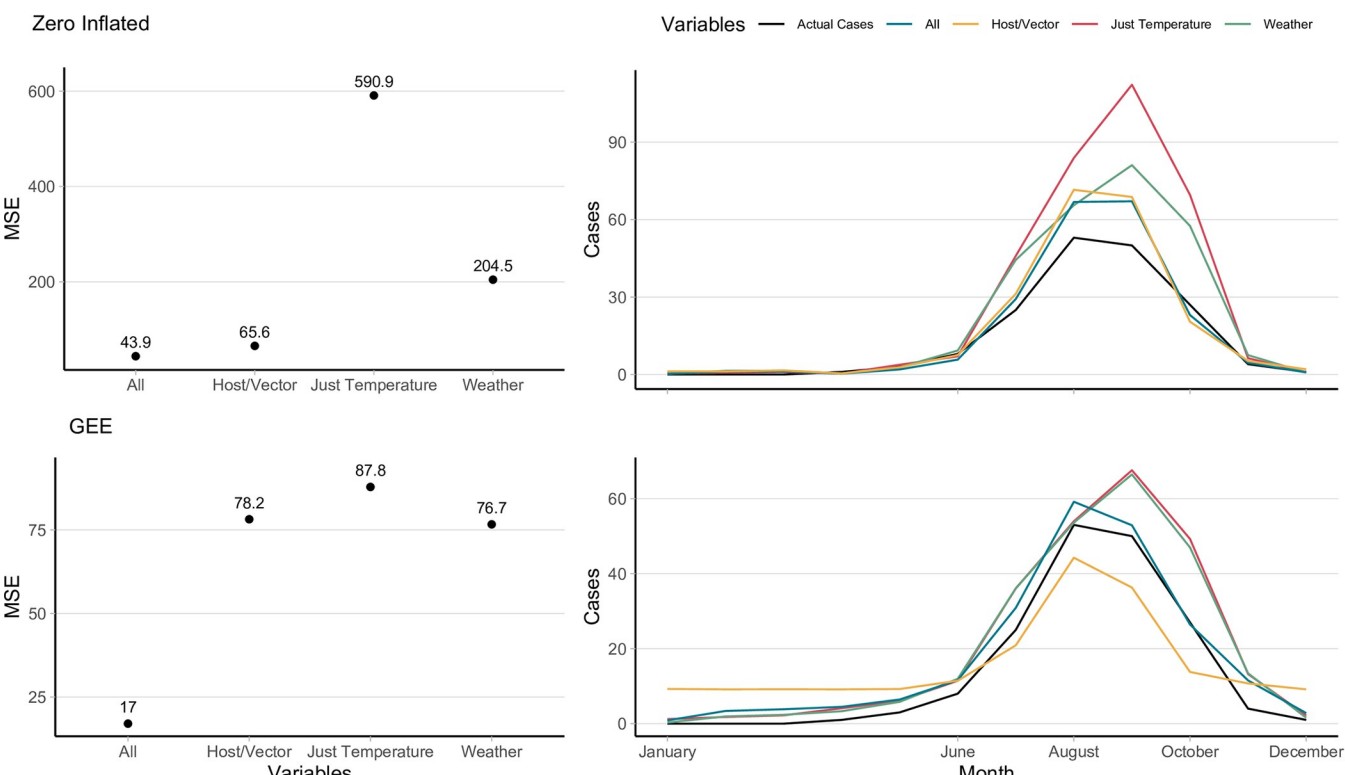

*\*All = Prior Month Temp, Prior Month Rainfall, Dead Birds, Mosquito Positivity Rate; Host/Vector = Dead Birds and Mosquito Positivity Rate; Weather = Prior Month Temp and Prior Month Rainfall*

**Fig 6. Predicted cases for 2022 for zero-inflated model and GEE model using different sets of variables.** Zero-inflated predictions on top row. GEE predictions on bottom row. A lower mean squared error (MSE) indicates a better fit.

Though it had the lowest MSE, the "All" GEE model still predicted more total cases in 2022 than were actually reported by a count of 214 to 172, respectively. The best performing model in this regard was the skewed "Host/Vector" GEE model (192 cases), followed by the "All" zero-inflated model (202 cases), and the "Host/Vector" zero-inflated model (214 cases).

In a simpler view, Fig 7 shows both the zero-inflated and GEE "All" models—the two best performing models according to MSE—plotted against the actual reported cases in the 11-high burden counties in 2022. This more clearly shows the differences between the prediction models discussed above.

## Discussion

Our study highlights an association between temperature and West Nile virus cases in 11 high-burden California counties from 2017–2022; holding month of year, location, and prior month rainfall constant, a one unit increase in prior month temperature was associated with 1.05 times the incidence of WNV. Given a hypothetical two degree Fahrenheit increase in average monthly temperature, which has been predicted for California in the next few decades due to climate change [17,18], the zero-inflated model predicted 24.8 excess cases per year over our study period. In the prediction models, using all variables—weather and host/vector combined—resulted in the best fit, and lowest mean squared error, for 2022 case counts. The GEE approach had overall better fitting prediction models, but the zero-inflated model was able to predict the total case count more closely for the year.

Our estimates on the significantly positive association between temperature and WNV incidence at the group level is well-supported in other studies in California and the US [9,20]. The

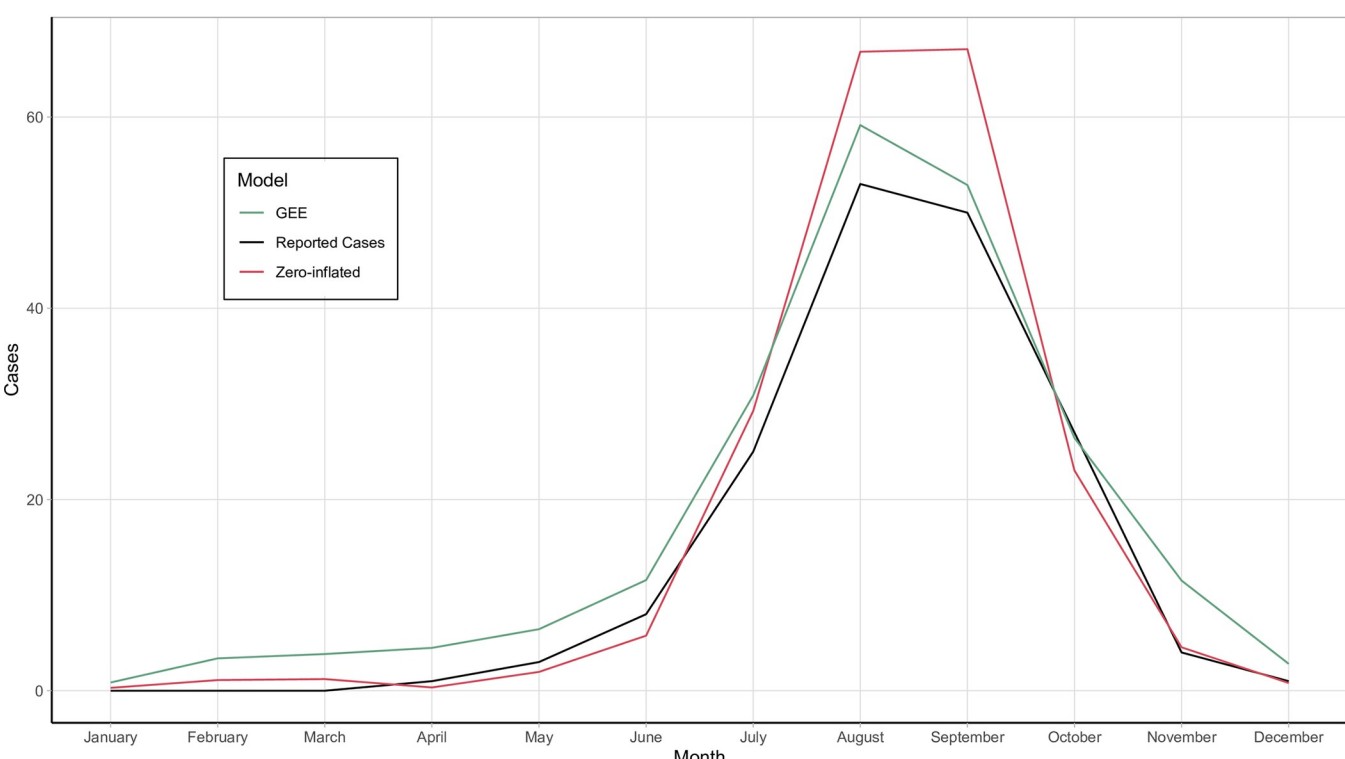

**Fig 7. Zero-inflated and GEE prediction models using "All" variables against reported cases in 11 high-burden California counties in 2022.**

magnitude of our association is in line with findings from Hahn et. al, who observed an odds ratio of 1.1 nationally [20]; alternatively, Hernandez et. al calculated an odds ratio of 10.5 in California's San Joaquin Valley, though this finding did have a wide confidence interval [9]. The connection between temperature and mosquito life traits, [6] such as development time and survival, as well as mosquito and viral interaction, such as the extrinsic incubation period —the time it takes for an infected mosquito to become infective to hosts—have been documented in numerous other studies [7,21]. These associations are complex and can often vary at different temperatures: e.g. Culex development rates rise with temperature until a certain point, when increasing temperature then causes rising mortality [22]. While this paper doesn't go into the biological mechanisms that mediate this relationship, these previous findings provide a causal basis for the observed association.

Given the results of both types of prediction models, it appears that having weather data alone provides little predictive power for WNV cases in California. Host and vector data provides a good fit in the absence of weather data for the zero-inflated model, showing that simple enzootic data from local vector departments can be enough to give useful information on current and future WNV cases in humans. However, given how readily available weather data is, it appears unnecessary and uninformative to use a prediction model with just host and vector data. This mostly aligns with the results of previous work. Wimberly et. al shows that while meteorological and mosquito infection rate data can predict WNV cases better than just historical data, it takes the combination of the two to best predict future cases [8]. Davis et. al's prediction model also showed that meteorological and mosquito data together can effectively predict future WNV cases, but again, disagreed with our model that mosquito infection data alone does nearly as well as all the data [23].

Our study revealed the power of the zero-inflated model to correctly predict the number of cases in low transmission months compared to more traditional models. However, the utility of this feature is limited, as low transmission months are of much smaller worry to health departments than summer months. The more traditional GEE model followed the shape and case counts of the peak summer months much closer and may provide health departments with a more accurate picture of what may come month to month. Though it overpredicts the number of cases during the summer, the zero-inflated model more accurately predicted the total number of cases in 2022, highlighting its potential utility in helping the state plan on a macro level. These models will need to be tested on a larger geographic scale with more counties and with newer years of data to make any final statements on their ability to be used by health departments.

A great strength of the study is its potential to be easily interpretable and implementable to public health agencies and the public: the data was easily accessible, and the model has just a few parameters that can be quickly integrated. These models allow for analyses that don't require complex information of the many relationships that underlie the observed associations, including the mechanisms by which temperature is associated with incidence. It also lends itself to future use by public health departments, which don't need large amounts of data or computational power to predict WNV cases. We also utilized standardization to estimate a risk difference in cases between the observed data and a hypothetical California with increased temperatures of various amounts, an underutilized strategy that can provide crucial insights into the future of WNV transmission in the state. Though hypothetical scenarios, it provides more interpretable estimates (in number of excess cases per month) than an odds ratio, or even our calculated rate ratio.

The data in this study has inherent limitations, however. Cases were aggregated on a monthly level, and other variables were in turn averaged or summed to correspond to this time frame. The lack of granularity in case data may bias the association between temperature and incidence, especially considering that weather can be highly volatile, a characteristic that is lost when averaged over a monthly time frame. Surveillance data, which comprises the case counts, dead bird counts, and mosquito pool positivity rate, also carries its own shortcomings, as it misses most, if not all, of asymptomatic infections and may not be representative of the infection dynamics in the system as a whole. We also lost granularity and possible variation in weather patterns and transmission dynamics across counties by analyzing the counties together; future work would benefit by looking at the effect of temperature on WNV incidence within counties, possibly at the zip code or census tract level.

There are also limitations based on the way the models were constructed, especially the zero-inflated model. Based on data from the last decade, we assumed that the month of year was a quality predictor of whether there would be non-zero case counts. However, studies have shown that a rise in temperature can also shift the seasonality of WNV in California [24]. This change would challenge our assumption and possibly bias our estimate of the association between temperature and WNV. Sensitivity analyses can be run to determine if other predictors, or sets of predictors, would be better suited as the zero count predictor in the zero-inflated model.

While this aggregated ecological study has limitations, it provides a manageable and convenient study design using pre-existing surveillance data. As previously mentioned, it is mandatory to report WNV disease, creating a passive surveillance system; as a result, no new data on the outcome needed to be assessed. The type of exposure—environmental—also pointed to using an ecological study, as temperature is a group-level exposure. It not only would be hard to measure the temperature experienced by an individual in a cohort, but generally uninformative to attempt to discern how a change in temperature affects a singular person's risk of WNV

disease, which may be influenced by many different possible exposures, including type of job and socioeconomic status.

Therefore, we don't believe to be committing any ecological fallacy—inferring that the ecological association equals the individual association—as we don't attempt to determine or make assumptions about the effect of temperature on the individual risk of WNV disease. Our study simply examines the risk aggregated at the county level. This is likely more informative to Californian public health agencies, as the risk of WNV disease to the average Californian is very low. In all, an ecological study was sufficient and better applicable to calculate the association of interest at the county level, while remaining replicable to future researchers and public health departments.

More types of climatic analysis can be conducted using this same data, such as the effect of prior seasonal rainfall and temperature on future WNV incidence, the importance of which has been shown for the length of the WNV season in the US as a whole [25,26]. Exploring the mechanisms that mediate the association between temperature and West Nile virus would also be beneficial to further analyze different intervention strategies. Finally, more work is required to understand the interface between humans and the enzootic carriers of WNV in California, and how climate change may alter the ways people interact and are exposed to these hosts and vectors.

## Conclusion

Increasing temperatures are associated with a higher incidence of West Nile virus infection in 11 high-burden California counties. Along with mosquito and bird infection data, meteorological data can closely predict future WNV incidence in these counties. This will be of extreme importance to public health agencies as WNV continues to become a larger health burden due to global warming and climate change.

## Acknowledgments

I would like to thank Dr. John Colford for his continued help with the structure, revisions, and publishing process of this work, Dr. Mary Beth Danforth for her guidance with the CDPH data, and Lincoln Wells and the team at VectorSurv for providing the host and vector data. This work was non-financially supported by the Department of Epidemiology at the Berkeley School of Public Health and was only possible with the assistance of numerous researchers, mentors, and peers at Berkeley.

## Author Contributions

**Conceptualization:** Noah Parker.

**Data curation:** Noah Parker.

**Formal analysis:** Noah Parker.

**Funding acquisition:** Noah Parker.

**Investigation:** Noah Parker.

**Methodology:** Noah Parker.

**Project administration:** Noah Parker.

**Resources:** Noah Parker.

**Software:** Noah Parker.

**Validation:** Noah Parker.

**Visualization:** Noah Parker.

**Writing – original draft:** Noah Parker.

**Writing – review & editing:** Noah Parker.

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
