## [Decision Letter · Decision Letter 0]

28 May 2024

Dear Mr. Parker,

Thank you very much for submitting your manuscript "Exploring the role of temperature and other environmental factors in West Nile virus incidence and prediction in California counties from 2017-2022" for consideration at PLOS Neglected Tropical Diseases. As with all papers reviewed by the journal, your manuscript was reviewed by members of the editorial board and by several independent reviewers. The reviewers appreciated the attention to an important topic. Based on the reviews, we are likely to accept this manuscript for publication, providing that you modify the manuscript according to the review recommendations. 

Your manuscript has been reviewed by several experts who considered it of high quality and valuable. One reviewer made some minor suggestions that might clarify some points in the manuscript. Please evaluate these, edit your manuscript appropriately, and re-submit it along with a brief explanation of the changes you made or did not make. Thank you!

Sincerely,

Richard A. Bowen

Academic Editor

Mabel Carabali

Section Editor

Your manuscript has been reviewed by several experts who considered it of high quality and valuable. One reviewer made some minor suggestions that might clarify some points in the manuscript. Please evaluate these, edit your manuscript appropriately, and re-submit it along with a brief explanation of the changes you made or did not make. Thank you!

Reviewer's Responses to Questions

**Key Review Criteria Required for Acceptance?**

**Methods**

-Are the objectives of the study clearly articulated with a clear testable hypothesis stated?

-Is the study design appropriate to address the stated objectives?

-Is the population clearly described and appropriate for the hypothesis being tested?

-Is the sample size sufficient to ensure adequate power to address the hypothesis being tested?

-Were correct statistical analysis used to support conclusions?

-Are there concerns about ethical or regulatory requirements being met?

Reviewer #1: The objectives of the study are clearly articulated with a clear testable hypothesis stated.

The study design is appropriate to address the stated objectives.

The population is clearly described and appropriate for the hypothesis being tested.

The sample size should be sufficient to ensure adequate power to address the hypothesis being tested. The authors could consider conducting a power analysis to confirm.

Correct statistical analysis methods were used to support conclusions.

There are no concerns about ethical or regulatory requirements being met.

Reviewer #2: -Are the objectives of the study clearly articulated with a clear testable hypothesis stated?

YES

-Is the study design appropriate to address the stated objectives?

YES

-Is the population clearly described and appropriate for the hypothesis being tested?

YES but some of the limitations and restrictions of the data seemed arbitrary and may need greater justification

-Is the sample size sufficient to ensure adequate power to address the hypothesis being tested?

YES

-Were correct statistical analysis used to support conclusions?

YES, probably but I am not a statistician and some terms were not sufficiently explained in common english. Further explanation is needed.

-Are there concerns about ethical or regulatory requirements being met?

NO

**Results**

-Does the analysis presented match the analysis plan?

-Are the results clearly and completely presented?

-Are the figures (Tables, Images) of sufficient quality for clarity?

Reviewer #1: The analysis presented matches the analysis plan.

The results are clearly and completely presented.

The figures and tables are of sufficient quality for clarity. 

In Table 1, however, the authors should consider clarifying what is meant by “average” here? Does it make more sense to report a mean with standard deviation or median with IQR based on the distribution of the data? Some measure of spread is needed here.

Reviewer #2: -Does the analysis presented match the analysis plan?

YES

-Are the results clearly and completely presented?

YES

-Are the figures (Tables, Images) of sufficient quality for clarity?

YES

**Conclusions**

-Are the conclusions supported by the data presented?

-Are the limitations of analysis clearly described?

-Do the authors discuss how these data can be helpful to advance our understanding of the topic under study?

-Is public health relevance addressed?

Reviewer #1: The conclusions are supported by the data presented.

The limitations of analysis are clearly described.

The authors discuss how these data can be helpful to advance our understanding of West Nile Virus incidence.

Public health relevance is addressed.

Reviewer #2: -Are the conclusions supported by the data presented?

YES

-Are the limitations of analysis clearly described?

YES

-Do the authors discuss how these data can be helpful to advance our understanding of the topic under study?

YES

-Is public health relevance addressed?

YES

**Editorial and Data Presentation Modifications?**

Reviewer #1: (No Response)

Reviewer #2: GEE and zero-inflated models are difficult concepts not clearly explained for a general reader.

Please give some common english explanations for these two terms that could help us to understand what they are and why they are different/more or less appropriate.

**Summary and General Comments**

Reviewer #1: The paper is well written, and the statistical methods are appropriate. The authors clearly explain and present a thorough discussion of the public health relevance.

Reviewer #2: In this manuscript, West Nile Virus notifications and other surveillance information are combined with rainfall and temperature to model existing real human notification data and then explore the effect of a two degree F mean monthly temperature increase on future human West Nile Virus cases in high transmission regions of California. The approach is generally interesting and potentially useful. As a non-statistician I would like more and clearer explanations of GEE and zero inflated models, what they are and how to understand the difference.

Please also consider the following questions that came to mind while reading the manuscript

1) how sensitive are your models to the assumptions you have made (restricting data examined to just 6 years) ?

2) did you explore other temperature scenarios besides 2 degrees? It might have made an interesting addition to look at 1, 1.5, 2 2.5 and 3 (for example) and see if a linear or non linear relationship emerges

3) could you consider monthly humidity data which may also effect mosquito survival?

4) some language is not quite scientific and should be revised to be more neutral eg : "mountains of data" "incredibly important"

PLOS authors have the option to publish the peer review history of their article (what does this mean?). If published, this will include your full peer review and any attached files.

Reviewer #1: No

Reviewer #2: No

Figure Files:

Data Requirements:

Reproducibility:

References

---

## [Editor Report · Decision Letter 1]

7 Jun 2024

Dear Mr. Parker,

We are pleased to inform you that your manuscript 'Exploring the role of temperature and other environmental factors in West Nile virus incidence and prediction in California counties from 2017-2022 using a zero-inflated model' has been provisionally accepted for publication in PLOS Neglected Tropical Diseases.

Best regards,

Richard A. Bowen

Academic Editor

Mabel Carabali

Section Editor

Thank you for the thoughtful revisions of your manuscript. I believe the modifications you made based on reviewer comments has improved your submission.

---

## [Editor Report · Acceptance letter]

18 Jun 2024

Dear Mr. Parker,

We are delighted to inform you that your manuscript, "Exploring the role of temperature and other environmental factors in West Nile virus incidence and prediction in California counties from 2017-2022 using a zero-inflated model," has been formally accepted for publication in PLOS Neglected Tropical Diseases.

Best regards,

Shaden Kamhawi

co-Editor-in-Chief

Paul Brindley

co-Editor-in-Chief
